# Brief Training of Technical Bleeding Control Skills—A Pilot Study with Security Forces

**DOI:** 10.3390/ijerph20032494

**Published:** 2023-01-31

**Authors:** Jose Luis Manteiga-Urbón, Felipe Fernández-Méndez, Martín Otero-Agra, María Fernández-Méndez, Myriam Santos-Folgar, Esther Insa-Calderon, María Sobrido-Prieto, Roberto Barcala-Furelos, Santiago Martínez-Isasi

**Affiliations:** 1REMOSS Research Group, University of Vigo, 36005 Pontevedra, Spain; 2School of Nursing, University of Vigo, 36005 Pontevedra, Spain; 3ESIMar (Mar Nursing School), Parc de Salut Mar, Universitat Pompeu Fabra Affiliated, 08003 Barcelona, Spain; 4SDHEd (Social Determinants and Health Education Research Group), IMIM (Hospital del Mar Medical Research Institute), 08003 Barcelona, Spain; 5Departamento de Ciencias da Saúde, Universidade de A Coruña (UDC), Campus de Esteiro, 15403 Ferrol, Spain; 6Simulation and Intensive Care Unit of Santiago (SICRUS) Research Group, Health Research Institute of Santiago, University Hospital of Santiago de Compostela-CHUS, 15706 Santiago Compostela, Spain; 7Faculty of Education and Sport Sciences, University of Vigo, 36005 Pontevedra, Spain; 8CLINURSID Research Group, Psychiatry, Radiology, Public Health, Nursing and Medicine Department, Universidade de Santiago de Compostela, 15782 Santiago de Compostela, Spain; 9Faculty of Nursing, Universidade de Santiago de Compostela, 15782 Santiago de Compostela, Spain

**Keywords:** public health, bleeding/prevention and control, extremity injury, tourniquet, law enforcement, learning

## Abstract

Uncontrolled external bleeding is a common cause of preventable death, and due to the environment in which these events often occur, e.g., in hostile environments, the state security forces are usually the first responders, and in many cases, if they are injured their partners provide the initial assistance. The tourniquet is a fast, effective, and easy-to-learn intervention, although there is a knowledge gap concerning training techniques. The objective is to evaluate the effectiveness of a bleeding control training program on a high-fidelity mannequin in a simulated critical situation in a law enforcement training environment. A quasi-experimental study was carried out with 27 members of the state security forces. They underwent brief theoretical–practical training and were evaluated via a scenario involving a critically ill patient in a hostile environment. The results showed that no member of the state security forces completed all the tourniquet placement steps, 26 (96%) prepared the tourniquet correctly, 21 (77.8%) placed it on the leg, and all the participants adjusted the band to the thickness of the injured limb and secured the windlass to the triangular flange of the device. However, only 23 (85.2%) of the participants placed it effectively. The participants, who were members of the state security forces, were able to effectively resolve a critical situation with active bleeding in a simulation scenario with a high-fidelity mannequin after completing theoretical–practical training.

## 1. Introduction

Uncontrolled external bleeding is a common cause of preventable death and is increasingly recognized as a serious public health problem [1,2,3].

In recent years, in attacks by active shooters and terrorists with knives and explosive devices, techniques learned in the military have been implemented in the civilian field because of the nature of the injuries. One of the main measures has been to treat uncontrolled external bleeding at the scene of the incident [4,5] by training the civilian population [5,6] and professionals such as police [7], health personnel [8], military [4], etc. The state security forces suffer 20% of attacks [9] and are the first responders in hostile environments and, in many cases, provide the initial medical assistance to civilians [10] or their partners.

The tourniquet is a resource with multiple benefits taken from tactical military medicine. In recent years, guidelines [11,12], action algorithms, and training plans have been developed [13,14,15], and the AHA [12] and the ERC [11] consider it an effective tool for the control of severe hemorrhage. It is a quick tool that can be applied in 60 to 80 s [16].

Despite the importance of this technique and the need for professionals who have mastered the use of the tourniquet, in 2018 [17], the International Liaison Committee on Resuscitation (ILCOR) established that there is a knowledge gap concerning the appropriate training techniques practiced by first aid providers. In 2020, it became clear that there was an urgent need to determine the educational requirements in order to develop an extensive strategy for training in the use and placement of tourniquets [9].

Therefore, the aim of this study is to evaluate the feasibility of a training program for the acquisition of technical skills in bleeding control on a high-fidelity mannequin in a simulated critical situation in a police training environment. The hypothesis put forward is that study participants would perform control of massive bleeding effectively and quickly after brief technical skills training.

## 2. Materials and Methods

### 2.1. Sample

A convenience sample of 27 members in active service in the state security forces (i.e., the National Police Corps, the Local Police, and the Civil Guard) participated in this study in 2018. They were stationed in the province of León in northern Spain. Their participation was voluntary and benevolent.

### 2.2. Study Design

This pilot study had a quasi-experimental design simulation without a control group in a simulation environment and was carried out to evaluate the abilities and sequence of action in the control of hemorrhage. The main outcomes of this study were: effective tourniquet placement (defined in variables such as “Effective tourniquet placement”) and rapid tourniquet control (performed in less than 60 s).

### 2.3. Training, Clinical Case, and Simulation Scenario

The participants received 90 min of theoretical and practical instruction on how to deal with massive hemorrhage and the use of the tourniquet. The training was divided into 45 min of theoretical presentation and 45 min of practical training.

The practical training consisted of an initial familiarization with the hemorrhage kit and training in pairs for the placement of a tourniquet on the upper and lower limbs.

A multidisciplinary group of medical experts, emergency care nurses, and members of the state security forces who specialize in simulation designed a clinical case with a patient in a critical situation. The simulation was based on the TCCC guidelines and only focused on medical aspects.

Prior to entering the scene, a first aid kit was placed on the work belt of each participant. The first aid kit was the same one used during the theoretical–practical training. The placement site, either on the left- or right-hand side, was chosen by each subject based on their comfort. The compact first aid kit for the control of hemorrhages (measuring 15 × 10 × 7 cm, with a capacity of 1.09 L and weighing 1350 g) included a SOF^®^ Tactical Tourniquet Wide tourniquet (SOF^®^TT-W) (Tactical Medical Solutions Inc., Anderson, SC, USA), Israeli compression bandage, 4″, Celox granulated hemostatic agent (35 g), 4 large-size vinyl gloves, and scissors for cutting clothes.

The assessment was conducted during tactical training exercises in regulation uniforms with simulated weapons. Participants were presented with the following scenario in a separate room: “A terrorist attack has just occurred with the detonation of an explosive. There is a victim inside and the scene is already under control. The terrorist has been killed. The scene is safe. You have to assess the victim”.

Upon entering the room, the participant found a METIman^®^ simulator (CAE Inc., Montreal, QC, Canada) programmed to suffer from an impaired level of consciousness, tachypnea, tachycardia, and hypotension. The simulator was lying on the ground and had active arterial bleeding. To simulate bleeding, the Xtreme Trauma BleedingLeg module (Simulaids Inc., Saugerties, NY, USA) was used with a partial amputation below the right lower extremity.

The scenario was followed by two expert instructors and recorded for later analysis using a Sony a6000 camcorder (Sony Corporation, Tokyo, Japan) for further study and analysis.

### 2.4. Variables

Demographic data, including sex, age, height, and weight, were recorded. Next, work and educational data (i.e., profession, years in the service, and previous training in hemorrhage control) were recorded.

Different variables were collected to assess bleeding control skills (Table 1).

### 2.5. Statistical Analise

Statistical analysis was performed with IBM SPSS Statistics software for Windows version 20 (SPSS Inc., IBM, Armonk, NY, USA).

In the analysis of quantitative data, we assessed the normality of the distribution by means of the Shapiro–Wilk test. We summarised quantitative variables using central tendency and dispersion measures (mean and standard deviation [SD]) and qualitative variables as absolute and relative frequency distributions. The association between categorical variables was assessed by means of the chi-square test.

## 3. Results

### 3.1. Characteristics of the Participants

The initial number of professionals from the state security forces was 27, 26 (96.3%) of whom were men, with a mean age of 40 ± 8 years.

Of the participants, 6 (22.2%) had received previous theoretical training; of these, 4 (66.7%) had received it in the last 12 months. The remaining 21 (77.7%) participants were not trained in tourniquet use prior to the study.

The participants were employed by the Civil Guard, 21 (78%), the Local Police, 4 (15%), and the National Police Corps, 2 (7%). The mean number of years practicing their profession was 18 ± 9 years.

### 3.2. Skill in Hemorrhage Control and Tourniquet Placement Time

Table 2 shows the results of tourniquet placement.

Once they accessed the simulation scenario, 96% (26) selected the tourniquet as the first treatment option, and 4% (1) applied direct pressure.

The most frequent site of tourniquet placement was the leg for 21 participants (77.8%), the knee for 4 (14.8%), and the other 2 (7.4%) chose the thigh.

With regard to the different skills required to correctly apply a tourniquet, all the participants were observed adjusting the band to the thickness of the injured limb and securing the windlass to the device’s triangular flange; 26 (96%) performed the 3 rotations of the windlass, and 18 (67%) marked the time of the placement. Less than half of the participants put on gloves, placed the tourniquet at the correct distance (between 5 and 7 cm), marked the time of placement, and checked the pulse.

Regarding the tourniquet placement distance, it was observed that 5 (18.5%) participants placed it properly (between 5 and 7 cm), and 22 (81.5%) placed it more than 7 cm away.

All participants achieved control of bleeding in less than 60 s (mean time 13.6 ± 7.3 cm).

### 3.3. Effective Tourniquet Placement

A total of 23 (85.2%) participants effectively placed the tourniquet in such a way that even if they did not follow all the steps of the protocol, its placement was able to stop a hemorrhage (Figure 1).

### 3.4. Proper Placement in Accordance with the Protocol

None of the study participants were able to complete all the steps of the per-protocol tourniquet placement (Figure 2). The errors that the participants made the most, in terms of proper placement according to the protocol, were the placement distance and checking the distal pulse.

No significant differences were observed when analyzing the influence of reported prior training on the TQ placement of participants with previous training and those who did not report prior training (*p* > 0.05).

## 4. Discussion

The objective of our study focused on evaluating the efficacy of a training program on hemorrhage control given to the state security forces and its evaluation through simulation.

All the participants adjusted the band to the thickness of the injured limb and secured the windlass to the triangular flange of the device.

The increase in incidents with mass casualties [5,18,19,20] has made it common for state security forces to be the first responders in emergency situations. In addition, the profile of their actions has varied due to the use of firearms and/or violent situations. Therefore, they must be trained in first aid, specifically in situations of massive bleeding, since uncontrolled bleeding continues to be the main cause of death in 35% of trauma patients [14].

The theoretical–practical training used in our study is one of the most used methodologies in training the use of tourniquets through the training of practical skills using feedback [21]. The training time is the same one used in bleeding control (B-CON) [21,22] training with good effectiveness results [21] and a tactical version provided by the TCCC [15] guidelines.

The participants selected the tourniquet as the first option in the treatment of massive hemorrhage. This may be because they identified a life-threatening situation and executed the most effective and appropriate technique for that specific profile of the injured person. This could be due to their work experience and may be a consequence of the unsafe context or the danger of their work, which conditions them in the application of direct pressure and compression, as it is difficult to carry out and maintain.

Serious penetrating injuries can cause life-threatening blood loss in less than 5 min [23,24]. Delays in clinical intervention are due to the response time of the emergency services and personnel restrictions related to security at the scene [3,23]. For this reason, state security forces are the ideal first responders in these situations. The participants in our study applied the tourniquet quickly for 24.5 ± 12.6 s, thus achieving control of early bleeding.

The decision made by the participants is effective because the early placement of the tourniquet before the onset of shock considerably increases survival [25,26,27]. Furthermore, the risk of delay due to applying other techniques is always high and even more so when the indication is clear.

Regarding the placement of the tourniquet and its location, 18.5% did so at a distance recommended by the TCCC and the European Resuscitation Council Guidelines for Resuscitation (5–7 cm) [11,28] with a tendency to move further away from the lesion (13.6 cm ± 7.3).

Using the most proximal tourniquet on the thigh would not decrease effectiveness, although it could condition the placement of the second tourniquet alongside it [25]. These results could be conditioned by the location of the amputation on the mannequin.

Our study evaluated efficiency by the suppression of active bleeding, which was achieved by performing three 180° rotations of the windlass [28]. If the tourniquet is fitted correctly to the thigh (100% of the participants fitted it properly), it will stop the blood flow. The use of the number of rotations of the windlass has been described as an effective measure for the suppression of active bleeding [25], although there is some controversy concerning that. The study by Childers R et al. found that the efficacy of 3.5 rotations was 60% [29].

The data from our study regarding the execution time showed that all the participants completed it in less than 60 s. Furthermore, 88.9% (24 subjects) completed it in less than 40 s, which is an excellent time [30,31,32].

These times are similar to those obtained by health professionals [16] and civilian personnel with similar training [24]. However, some details have an influence on the final placement time and the comparison of times between studies. These include factors such as access to the tourniquet, how the device is attached to the belt, whether it is with a button or a hook closure system and a loop, or how the tourniquet is stored [33].

Skills assessment was performed using the high-fidelity simulator with active arterial bleeding. Its use improved the realism of the scene and, based on previous studies, it improves the evaluation since it takes less time to control the hemorrhage, the patient loses less blood, and the student has more confidence when executing the technique [28].

Our study has multiple limitations, the main one being the difficulty of recreating a real scenario in a simulated scenario. Performing a tourniquet effectively in a stressful situation in an adequate time is more complex, although, in the study by Tsur et al. [34], better results were obtained when stressful factors affected the participants.

A limitation of the simulator is that it does not measure arterial occlusion pressure. To overcome this limitation, we selected three rotations of the windlass as a condition for arterial occlusion [28].

Another limitation is the type of tourniquet used. According to the study by Schreckengaust et al. [30], the placement of the SOFFT tourniquet produces worse results. Even so, our results are good, and it is assumed that the results will be better with another type of tourniquet and prior training. It is also necessary to mention that the sample size used in this study is not representative, and furthermore, the methodology and the unavailability of a control group limit the interpretation of the results.

## 5. Conclusions

In conclusion, the participants, members of the state security forces, were able to effectively resolve a critical situation with active bleeding in a simulation scenario with a high-fidelity mannequin after theoretical–practical training. However, other simulation training techniques or new technologies that could offer other results in terms of application according to the protocol should be explored.

## Figures and Tables

**Figure 1 ijerph-20-02494-f001:**
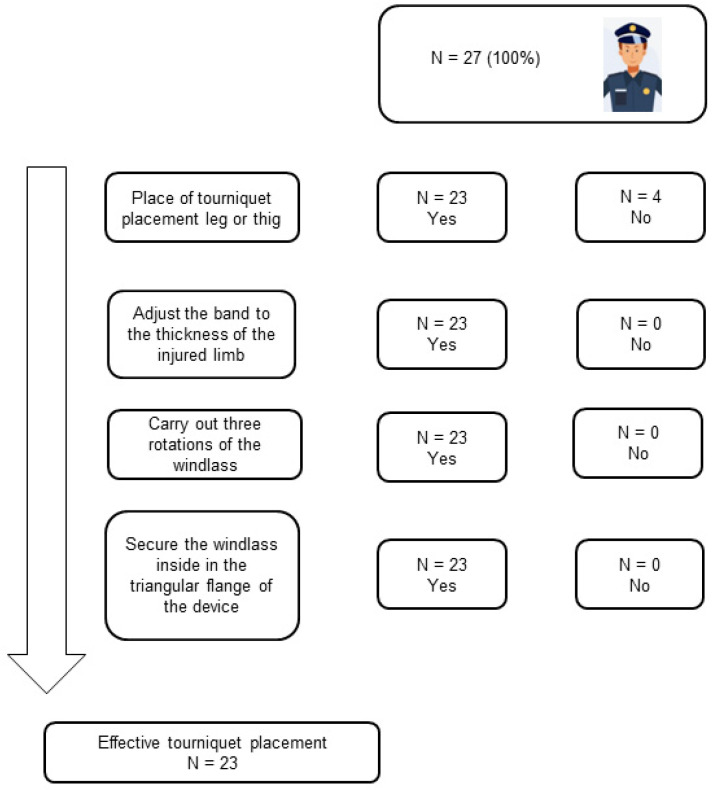
Effective Tourniquet Placement Flow Chart.

**Figure 2 ijerph-20-02494-f002:**
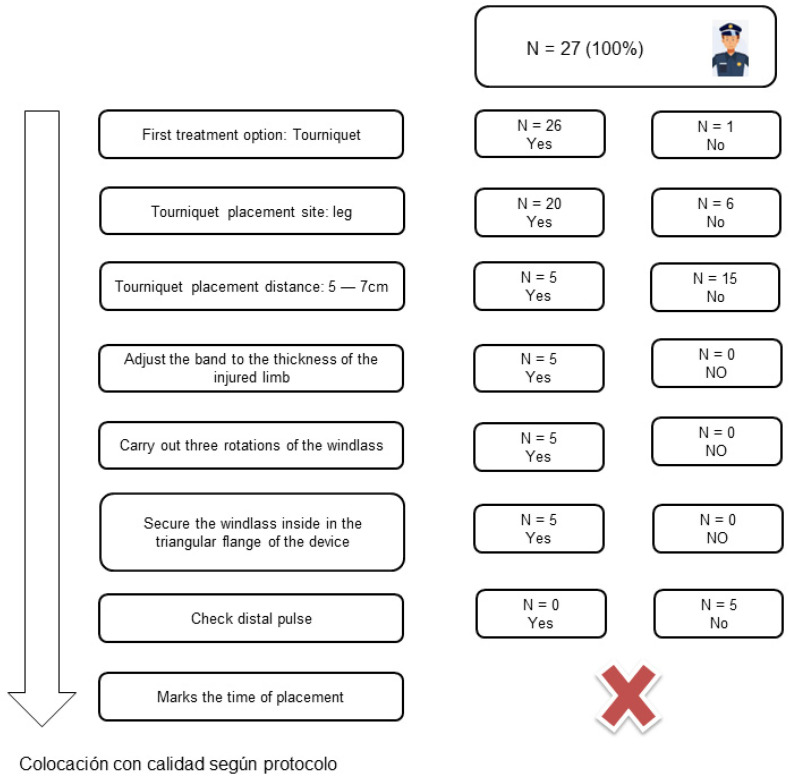
Proper placement flow chart in accordance with the protocol.

**Table 1 ijerph-20-02494-t001:** Skill variables in hemorrhage control.

Action	Options	Explanation
Putting on gloves	Yes	
	No	
What is the first option in the treatment of hemorrhage?	Tourniquet	It is considered the first technique you use, even if you end up using a different one later
	Direct pressure	
	Hemostatic agents	
	Compression bandage	
Adjust the band to the thickness of the injured limb before rotating the windlass	Yes	It is considered a good fit when it is not possible to insert the tips of three fingers transversely between the tourniquet and the patient’s limb [12]
	No	
Carry out three rotations of the windlass	Yes	The tourniquet should be applied around the extremity in three 180-degree rotations (540-degree total rotation) [12]
	No	
Secure the windlass inside the triangular flange of the device	Yes	
	No	
Do you check the distal pulse after the bleeding stops?	Yes	Consider taking a pulse by palpation at any site below the tourniquet
	No	
Mark the time at which you place the tourniquet	Yes	
	No	
Site of tourniquet placement	Leg	
	Knee	
	Thigh	
Correct distance placement		In centimeters
How far from the wound is the tourniquet placed?		Distance measured in cm from the wound to the tourniquet. The correct distance is considered between 5 and 7 cm above the lesion [7,12]
Tourniquet placement time		Time from when the subject takes the tourniquet in his hand until he finishes placing it (in seconds)
Effective tourniquet placement	Effective	Effective placement is considered to be when it is effective in controlling bleeding, even if the protocol has not been followed exactly. To achieve this, the subject must place the tourniquet (on the leg or thigh), adjust the band to the thickness of the injured limb, perform 3 rotations of the windlass, and secure the windlass in the triangular flange of the device.
	Ineffective	A placement is considered ineffective when it does not accomplish any of the effective placement steps.
Proper placement in accordance with the protocol	Correct	Correct placement is considered to be when all the steps have been carried out in the indicated order. To achieve this, the subject must place the tourniquet on the leg at a distance of about 5–7 cm, adjust the band to the thickness of the injured limb, perform 3 rotations of the windlass, secure the windlass in the flange, check the distal pulse, and finally, mark the placement time.
	Wrong	Incorrect placement is considered to be when all the steps followed in the correct placement are not carried out or are not carried out in the proper order.

**Table 2 ijerph-20-02494-t002:** Skill Results.

Variables	Result	Unit of Measurement
Putting on gloves	12 (44.4)	Number of Participants (%)
What is the first option in the treatment of hemorrhage?	Tourniquet 26 (96.3%)	
	Direct pressure → 1 (3.7)	
Adjust the band to the thickness of the injured limb before rotating the windlass	27 (100)	
Rotate the windlass three times	26 (96.3)	
Secure the windlass into the triangular flange of the device	27 (100)	
Do you check the distal pulse after the bleeding stops?	1 (3.7)	
Mark the time at which you place the tourniquet	18 (66.7)	
Adjust the band to the thickness of the injured limb before rotating the windlass	27 (100)	
Site of tourniquet placement	Leg 21 (77.8)	
	Knee 4 (14.8)	
	Thigh 2 (7.4)	
Correct distance placement	5 (18.5)	
How far from the wound is the tourniquet placed	13.6 ± 7.3 cm	Mean ± standard deviation
Tourniquet placement time	24.5 ± 12.6 s	

## Data Availability

Not applicable.

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
