# Peer review of "Brief Training of Technical Bleeding Control Skills—A Pilot Study with Security Forces"

_ijerph, 2023, doi:10.3390/ijerph20032494_

Round 1

Reviewer 1 Report (Previous Reviewer 1)

The authors have provided adequate responses to the issues raised in the previous review report (ijerph-1958648).

Author Response

We would like to thank the reviewers for their detailed and constructive comments to improve our manuscript. We have attempted to make the changes accordingly covering all suggestions and comments in the revised version of the manuscript.

Reviewer 2 Report (Previous Reviewer 2)

This is a review of a paper resubmitted after a prior "reject and encourage resubmission" decision.

There have been some improvements since the last review round, but I do not believe that this paper can be published in its current state. THe main issue is that the rationale for this pilot study and for its more than limited sample size is unclear. The authors could choose to use the data they gathered to design a larger, randomized controlled trial, and this manuscript could therefore be converted either in a (very) brief report which would support the creation of a protocol, or directly in a study protocol.

Major comments

1. I am not convinced by the rationale for a pilot study in this setting. In the introduction, the authors state that "everyone agrees that the tourniquet is a fast, effective and easy-to-learn intervention method". While I am not convinced that this is truly the case (and the reference does not adequately support this statement either), I fail to understand why such a study would be useful if it were.

2. A greater attention should be given to references. In addition to the debatable use of references to support doubtful statements, some references are incomplete (for instance, ref 6 lacks a title among other information: Valiño EM,; Castro P,; Castro Delgado R. Emergencias. 2022, p. :458-64.). After some research, I found the actual reference, entitled "Análisis descriptivo de los incidentes con múltiples víctimas intencionados en entorno civil en Europa durante el periodo 2000-2018". This reference describes the occurence of intentional mass casualty incidence in Europe, and shows that explosions and firearm attacks are the main types of incidents. The words "tourniquet" (torniquete) or "hemorrhage" (hemorragia) do not even appear in the manuscript. Using this reference to support the sentence "In 2020, it became clear that there was an urgent need to determine the educational requirements in order to develop an extensive strategy for training in the use and placement of tourniquets" is therefore, at best, questionable.

3. Methods: In their reponse to prior review comments, the authors indicate that they used the CONSORT guidelines to report their study, but no guideline is currently acknowledged in this section. The appropriate guidelines should nevertheless be acknowledged and the fully completed checklist should be provided as supplementary material. In addition, the sentence "This pilot study had a A quasi-experimental desing of controlled simulation without a control group" is confusing: using "control" and "controlled" for two different aspects in the same sentence should be avoided.

4. Furthermore, while an objective is indeed stated ("the objective of this study was to evaluate the effectiveness of a bleeding

control training program"), it is rather broad and there is no hypothesis as to what the training program is supposed to achieve. In addition, no outcomes have been defined, and the word "outcome" does not even appear in the manuscript. The data gathered through this study could be used to calculate a sample size for a larger, randomized controlled trial.

5. Statistical analysis: while the authors state that they "summarised quantitative variables using central tendency and dispersion measures (mean/median and standard deviation [SD]/interquartile range [IQR]), I have been unable to find even a single median or IQR in the results.

6. The limitations section should be revised: the main limitations of this study are the small sample size and the lack of a control group.

Minor comments

7. English editing: there are still a few typos ("This pilot study had a A quasi-experimental [...]) and tenses are not always consistent ("During tactical training exercises with regulation uniform and simulated weapons. They are interrupted and presented with a scenario in a separate room. Each participant was then presented with the following scenario").

Author Response

Manuscript entitled " Brief training of technical bleeding control skills.  A pilot study with security forces”

We would like to thank the reviewers for their detailed and constructive comments to improve our manuscript. We have attempted to make the changes accordingly covering all suggestions and comments in the revised version of the manuscript.

COMMENTS:

Reviewer 2

Major comments

  1. I am not convinced by the rationale for a pilot study in this setting. In the introduction, the authors state that "everyone agrees that the tourniquet is a fast, effective and easy-to-learn intervention method". While I am not convinced that this is truly the case (and the reference does not adequately support this statement either), I fail to understand why such a study would be useful if it were.
    1. We appreciate the reviewer's comments. We agree that the referenced sentence does not adequately support the statement. That is why it has been changed to "tourniquet is an effective intervention method". We believe that this study provides evidence, in line with the study design, in relation to the description of a type of tourniquet training for law enforcement officers. We believe that with the results of this study we can add knowledge to prepare the training of security personnel.
  2. A greater attention should be given to references. In addition to the debatable use of references to support doubtful statements, some references are incomplete (for instance, ref 6 lacks a title among other information: Valiño EM,; Castro P,; Castro Delgado R. Emergencias. 2022, p. :458-64.). After some research, I found the actual reference, entitled "Análisis descriptivo de los incidentes con múltiples víctimas intencionados en entorno civil en Europa durante el periodo 2000-2018". This reference describes the occurence of intentional mass casualty incidence in Europe, and shows that explosions and firearm attacks are the main types of incidents. The words "tourniquet" (torniquete) or "hemorrhage" (hemorragia) do not even appear in the manuscript. Using this reference to support the sentence "In 2020, it became clear that there was an urgent need to determine the educational requirements in order to develop an extensive strategy for training in the use and placement of tourniquets" is therefore, at best, questionable.
    1. We have changed the wording to make it clearer.
    2. “In recent years, attacks by active shooters and terrorists with knives and explosive devices and, and, because of the nature of the injuries, what has been learned in the military has been implemented in the civilian field. One of the main measures has been to treat uncontrolled external bleeding at the scene of the incident. [4,5], by training the civilian population [5][6] and professionals such as police [7], health personnel [8], military[4], etc). In this sense, the state security forces suffer 20% of attacks [9] and in addition, are the first responders in hostile environments and in many cases, to provide the initial medical assistance a civil persons[10] or their partners.

The tourniquet is a resource with multiple benefits taken from tactical military medicine. In recent years, guidelines[11,12], action algorithms and training plans have been developed in this regard[13–15], and the AHA[16] and the ERC[11] consider it an effective tool for the control of severe haemorrhage, and it is also a quick tool that can be applied in 60 to 80 seconds[17].

  1. The literature shows that in explosions and firearm attacks, massive haemorrhages are common and therefore, being hot and unsafe areas. The first responders are usually police officers.
  1. Methods: In their reponse to prior review comments, the authors indicate that they used the CONSORT guidelines to report their study, but no guideline is currently acknowledged in this section. The appropriate guidelines should nevertheless be acknowledged and the fully completed checklist should be provided as supplementary material.
    1. The reviewer's recommendation is accepted. We have attached the fully completed checklist as supplementary material to the submission. As this is a quasi-experimental design and not a randomised trial, there are some items in the checklist that do not appear in the manuscriptIn addition, the sentence "This pilot study had a A quasi-experimental desing of controlled simulation without a control group" is confusing: using "control" and "controlled" for two different aspects in the same sentence should be avoided.
    2. The reviewer's recommendation is accepted. We have changed it
  2. Furthermore, while an objective is indeed stated ("the objective of this study was to evaluate the effectiveness of a bleeding control training program"), it is rather broad and there is no hypothesis as to what the training program is supposed to achieve. In addition, no outcomes have been defined, and the word "outcome" does not even appear in the manuscript. The data gathered through this study could be used to calculate a sample size for a larger, randomized controlled trial.
    1. The reviewer's recommendation is accepted. We have changed “effectiveness” by “feasibility” in the objective. We have added in the text. We have also defined in the text the outcomes proposed
  3. Statistical analysis: while the authors state that they "summarised quantitative variables using central tendency and dispersion measures (mean and standard deviation [SD]/interquartile range [IQR]), I have been unable to find even a single median or IQR in the results.
    1. The reviewer's recommendation is accepted. We have changed it.
  4. The limitations section should be revised: the main limitations of this study are the small sample size and the lack of a control group.
    1. The reviewer's recommendation is accepted. We have explained in the limitations of the study everything related to the sample size, the type of methodology used and the lack of a control group.
  5. Minor comments
  6. English editing: there are still a few typos ("This pilot study had a A quasi-experimental [...]) and tenses are not always consistent ("During tactical training exercises with regulation uniform and simulated weapons. They are interrupted and presented with a scenario in a separate room. Each participant was then presented with the following scenario").
    1. we have changed it

This manuscript is a resubmission of an earlier submission. The following is a list of the peer review reports and author responses from that submission.

Round 1

Reviewer 1 Report

Thank you very much for the opportunity to review this manuscript, submitted by Manteiga et al.

It is a well-conceived work, focusing on the need for proper training in first aid for law enforcement. They are often the first to respond to serious trauma scenarios where effectively controlling the bleeding is vital. In fact, the World Health Organisation (WHO) recommends involving lay people in prehospital care. However, in order to increase the quality of this work, there are some aspects that need to be reconsidered.

1.     The introduction should provide more information about the benefits of implementing this type of program in basic first aid skills; and how its success depends on saving multiple lives of polytraumatized patients.

2.     Another issue to be better explained is the inclusion of the 27 people in the study; and the reason for not including a control group. It might be interesting to carry out a control group without previous training, to evaluate whether the previous theoretical-practical training was successful.

3.     The authors indicated that the participants received a 90-minute theoretical-practical course. Why did they not consider to evaluate the knowledge learned in this course? It may be that these results could explain some of the findings found in the subsequent scenario.

4.     They also stated that a multidisciplinary group designed the clinical scenario. Did any external validation of the scenario take place? Either by critical care physicians/nurses or by external clinical simulation specialists. And, if no external validation was performed, were any official or consensus guidelines followed for its elaboration?

5.     They set the clinical case in the context of a terrorist attack; did they incorporate any additional elements to make it more real? Noises, sirens, debris...

6.     Finally, concerning the statistical analyses, they should explain better why they used such descriptive statistics.  From the data provided, it could be possible that the variables did not fit a normal distribution, so expressing it as the mean and standard deviation is not correct. Could you please check and provide more data on this aspect?

7.     The results are written in a clear and orderly manner. Focusing on the main aspects of the study. As well as the discussion, which is appropriate for the results presented in this first version of the manuscript.

8.     The conclusions are appropriate, and directly arise from the data presented, although it should be made clear that this is a pilot test, with a very low number of participants. This aspect might also be reflected in the title of the manuscript.

Reviewer 2 Report

The authors have carried out a quasi-experimental study to assess the effectiveness of a training program on hemorrhage control among 27 members of state security forces. They found that no participant was able to complete all the tourniquet placement steps but concluded that they were nevertheless "able to effectively resolve a critical situation with active bleeding in a simulation scenario" using a METIman® simulator.

This is an important topic, but there are many limitations to this study, the most important of which is the very limited sample size and the lack of sample size calculation (there is little reason to use a convenience sample when participants come from "the National Police Corps, the Local Police and the Civil Guard").

The study design is questionable (a randomized controlled trial does not seem difficult to carry out in this setting and using such participants). The lack of a control group decreases the potential worth of this study.

Ideally, the study should be registered first, and a protocol should be designed prior to carrying out the study (using the SPIRIT guidelines). In addition, an appropriate reporting guideline should be used (the STROBE guidelines would be suitable for the current study).

Statistical analyses should be described further. Normality should be assessed prior to deciding whether to report means (SD) or median (IQR).

Finally, manuscripts should be revised by proficient English writers before submission.